# Precise synthesis of sulfur-containing polymers via cooperative dual organocatalysts with high activity

Cheng-Jian Zhang[1], Hai-Lin Wu[1], Yang Li[1], Jia-Liang Yang[1] & Xing-Hong Zhang [1]

Metal-free and controlled synthesis of sulfur-containing polymer is still a big challenge in polymer chemistry. Here, we report a metal-free, living copolymerization of carbonyl sulfide (COS) with epoxides via the cooperative catalysis of organic Lewis pairs including bases (e.g.: phosphazene, amidine, and guanidine) and thioureas as hydrogen-bond donors, afford well-defined poly(monothiocarbonate)s with 100% alternating degree, >99% tail-to-head content, controlled molecular weights (up to 98.4 kg/mol), and narrow molecular weight distributions (1.13–1.23). The effect of the types of Lewis pairs on the copolymerization of COS with several epoxides is investigated. The turnover frequencies (TOFs) of these Lewis pairs are as high as 112 h$^{-1}$ at 25 °C. Kinetic and mechanistic results suggest that the supramolecular specific recognition of thiourea to epoxide and base to COS promote the copolymerization cooperatively. This strategy provides commercially available Lewis pairs for metal-free synthesis of sulfur-containing polymers with precise structure.

[1] MOE Key Laboratory of Macromolecular Synthesis and Functionalization, Department of Polymer Science and Engineering, Zhejiang University, Hangzhou 310027, China. Correspondence and requests for materials should be addressed to X.-H.Z. (email: xhzhang@zju.edu.cn)

The finding of fresh monomers[1,2] and the development of active catalysts[3,4] are the central topics in synthetic polymer chemistry. Carbonyl sulfide (COS), a key intermediate of the atmospheric sulfur cycle and the most abundant sulfur-containing gas in the troposphere, causes haze, acid rain, and ozonosphere damage[5], and is also a one-carbon ($C_1$) heterocumulene and structural analog of carbon dioxide ($CO_2$). Utilizing COS to copolymerize with epoxides is a emerging atom-economic and versatile approach to produce functional sulfur-containing polymers[6–11]. In contrast, traditional synthesis of sulfur-containing polymers often involves the condensation of thiols with phosgene and ring-opening polymerization (ROP) of cyclic thiocarbonates that are generally derived from thiols and phosgene[6].

Recent synthetic advances[6–11] provide metal catalytic strategies for making a variety of COS-derived poly(monothiocarbonate)s that have good solubility in organic solvents, superior optical properties, and excellent chemical resistance[6,7]. Zinc-cobalt(III) double-metal cyanide complexes[8] and (salen)CrX/onium salts (Lewis bases, LBs, Fig. 1a)[9–11] have been discovered to catalyze the COS/epoxide copolymerization. Unfortunately, metal-contaminated or colored copolymers are resulted and severely impede their applications in optical, optoelectronic, photochemical, or biomedical materials[12–14]. Organic Lewis pairs composed of triethylborane (TEB) and LBs, including amidine, quinidine, quaternary onium salts could generate poly(monothiocarbonate)s from COS and epoxides[15] (Fig. 1b). However, the TEB/LB pairs often led to relative broad polydispersity (PDI) and higher molecular weights than the calculated one. Meanwhile, only TEB, which is toxic and spontaneous combustion in air, was effective to COS/epoxide copolymerization.

The above-mentioned catalysts have the coordination bonds that are responsible to the activation of the monomers. Whereas very few of reports suggested an anionic copolymerization process involved $C_1$ monomers. For example, Nozaki et al., disclosed that [PPN]Cl could solely catalyze the carbon-disulfide ($CS_2$)/propylene sulfide copolymerization[16]. Feng and Gnanou et al. presented that alkoxide/benzyl alcohol (BnOH) could effectively initiate the $CO_2$/epoxide copolymerization[17,18]. However, anionic copolymerization of COS with epoxides remains unexplored. In contrast with the $CO_2$/epoxide copolymerization that is often expected to attain fully alternating structure and no production of side cyclic carbonate (i.e., 100% polycarbonate)[19–24], the chemistry of COS/epoxide copolymerization is more complicated[6].

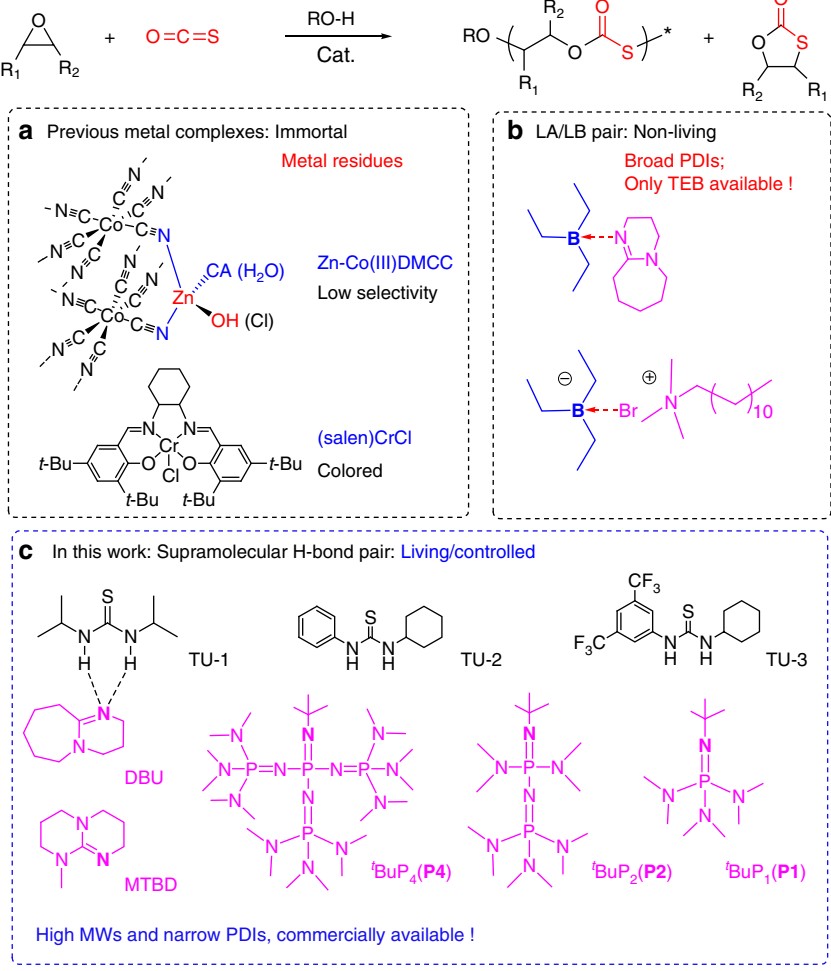

**Fig. 1** The catalyst systems for COS/epoxide copolymerization. **a** Metal catalysts (zinc-cobalt(III) double-metal cyanide complexes and (salen)CrX/onium salts), **b** TEB/LB pairs (triethylborane/Lewis bases), **c** TU/base pairs in this study (TU-1: diisopropyl thiourea; TU-2: 1-cyclohexyl-3-phenylthiourea; TU-3: 1-[3,5-bis(trifluoromethyl) phenyl]-3-cyclohexylthiourea; DBU: 8-diazabicyclo[5.4.0]undec-7-ene; MTBD: *N*-methyl-1,5,7-triazabicyclododecene; $^t$Bu-P$_4$: **P4**, 1-*tert*-butyl-4,4,4-tris(dimethylamino)-2,2-bis [tris(dimethylamino)- phosphoranylidenamino]-2λ5,4λ5-catenadi(phosphazene); $^t$Bu-P$_2$: **P2**, 1-*tert*-Butyl-2,2,4,4,4-pentakis(dimethylamino)-2λ5,4λ5-catenadi(phosphazene); and $^t$Bu-P$_1$: **P1**, *tert*-butylimino-tris(dimethylamino) phosphorene)

**Table 1 COS/PO copolymerization catalyzed by TU-1/organic base pairs at 25 °C**

| Entry[a] | LB | [PO]: [LB]: [TU-1]: [I] | TOF (h$^{-1}$)[b] | Copolymer selectivity[c] | Alternating degree (%)[c] | T–H diad content (%)[d] | O/S ER product[d] | $M_n$ (kg/mol)[e] | PDI[e] |
|---|---|---|---|---|---|---|---|---|---|
| 1 | DBU | 250:1:0:0 | 4 | 93/7 | 100 | >99 | N.F. | 38.3 | 1.35 |
| 2 | **P4** | 1000:1:0:0 | 23 | 60/40 | 92/8 | >99 | N.F. | 29.4 | 1.58 |
| 3 | **P2** | 250:1:0:0 | 4 | 98/2 | 100 | >99 | N.F. | 45.1 | 1.14 |
| 4 | **P1** | 250:1:0:0 | 1 | >99 | 100 | >99 | N.F. | 9.0 | 1.05 |
| 5 | MTBD | 250:1:0:0 | 3 | 96/4 | 100 | >99 | N.F. | 46.0 | 1.20 |
| 6 | DBU | 2000:1:0:1 | 10 | 92/8 | 100 | >99 | N.F. | 13.9 | 1.20 |
| 7 | **P4** | 2000:1:0:1 | 55 | 99/1 | 100 | >99 | N.F. | 42.5 | 1.23 |
| 8 | **P2** | 2000:1:0:1 | 29 | 99/1 | 100 | >99 | N.F. | 35.6 | 1.15 |
| 9 | **P4** | 2000:1:1:0 | 41 | 97/3 | 100 | >99 | N.F. | 31.3 | 1.16 |
| 10 | **P2** | 1000:1:1:0 | 24 | >99 | 100 | >99 | N.F. | 34.3 | 1.16 |
| 11 | DBU | 250:1:1:0 | 7 | 93/7 | 100 | >99 | N.F. | 30.2 | 1.13 |
| 12 | MTBD | 250:1:1:0 | 4 | 95/5 | 100 | >99 | N.F. | 36.1 | 1.16 |
| 13[f] | MTBD | 1000:1:1:0 | 10 | 98/2 | 100 | >99 | N.F. | 98.4 | 1.14 |
| 14 | **P4** | 4000:1:1:1 | 75 | >99 | 100 | >99 | N.F. | 21.3 | 1.21 |
| 15 | **P4** | 4000:1:5:5 | 112 | 98/2 | 100 | >99 | N.F. | 9.9 | 1.13 |
| 16 | **P2** | 2000:1:1:1 | 35 | >99 | 100 | >99 | N.F. | 19.3 | 1.13 |
| 17 | DBU | 2000:1:1:1 | 22 | 94/6 | 100 | >99 | N.F. | 11.3 | 1.19 |
| 18 | MTBD | 2000:1:1:1 | 17 | 97/3 | 100 | >99 | N.F. | 12.0 | 1.23 |

The copolymerization results under other conditions (including controls), representative NMR spectra and corresponding calculations are in Supplementary Figs. 2–11 and 35, and the Methods
[a]Reactions were run at 25 °C in neat PO (1.0 ml; COS: PO = 1.2: 1) in a 10 ml autoclave, 24 h
[b]Turnover of frequency (TOF), (Mol epoxide consumed)/(mol LB h), PO conversion was determined by $^1$H NMR spectroscopy
[c]Determined by $^1$H NMR spectroscopy. The copolymer selectivity is the molar ratio of the copolymer/cyclic product; The alternating degree is the molar percentage of the monothiocarbonate unit in the polymer chain
[d]Determined by $^{13}$C NMR spectroscopy. O/S ER = oxygen-sulfur exchange reaction. N.F. = not found (dithiocarbonate and carbonate units)
[e]Determined by gel permeation chromatography in THF, calibrated with polystyrene standards
[f]72 h

One is the possible occurrence of oxygen/sulfur exchange reactions (O/S ERs), which cause the production of $CO_2$, and thiirane intermediate, will produce randomly distributed dithiocarbonate and carbonate units in the final copolymer[8,25–27]. The other is that the copolymerization of structurally asymmetric COS with a terminated epoxide, will generate four consecutive monothiocarbonate diads, i.e.,: head-to-tail (H–T), tail-to-head (T–H), tail-to-tail (T–T), and head-to-head (H–H) diads[6]. As a result, metal-free catalyst for anionic COS/ epoxide copolymerization should avoid O/S ER, attain highly regioselectivity involved two asymmetric monomers and be precisely controlled by varying the monomer to initiator ratios under mild condition.

Herein, we have developed a living copolymerization of COS with various epoxides with high activity, using commonly available thioureas (TUs) and organic LBs (DBU, MTBD, **P4**, **P2**, and **P1**, Fig. 1c). This catalyst system was developed based on the hypothesis that the cooperative catalytic process of Lewis pairs composed of TU and base, undergoing a non-covalent mode to activate and stabilize the alcohol initiator/chain end for controlling the anionic copolymerization[28–31]. The resulting poly(monothiocarbonate)s have perfectly alternating structure with regioregularity, controlled molecular weights and narrow PDIs, and are colorless (Supplementary Fig. 1).

## Results

**Anionic COS/propylene oxide (PO) copolymerization.** Previous studies have shown that the organic bases, e.g.: amidine and guanidine widely employed as the cocatalysts for metal complex for catalyzing $CO_2(COS)$ /epoxide copolymerization[6,32–34]. In this scenario, the organic bases often exhibited no activity towards $CO_2$/epoxide copolymerizations[19–24,32–34]. Since COS is more reactive than $CO_2$ and the expected thiocarbonate anion [OC(=O)S$^-$] is more nucleophilic than carbonate anion [OC(=O)O$^-$][6], we performed the sole catalysis of LBs (DBU, MTBD, **P4**, **P2**, and **P1**) for COS/PO copolymerization as controls

(entries 1–5 in Table 1) for comparatively studying the catalytic performance of the designed Lewis pairs of TU/LB (entries 9–18 in Table 1). Unexpectedly, we observed that DBU could solely catalyze COS/PO copolymerization in a high PO/DBU feed ratio of 250/1 at ambient temperature (25 °C) for a long time of 24 h (entry 1, Table 1). Poly(propylene monothiocarbonate)s (PPMTCs) were obtained with a number-average molecular weight ($M_n$) of 38.3 kg/mol with production of 7% of cyclic monothiocarbonate. **P4**, a phosphazene with the strongest basicity[35], was active for the anionic ROP of epoxides, cyclic esters, and caprolactam[36–39], was also effective to the COS/PO copolymerization with a TOF of 23 h$^{-1}$ even under a PO/**P4** feed ratio (1000/1). However, the copolymer selectivity was only 60%, and 8% polyether was detected (entry 2 in Table 1, Supplementary Fig. 2). The sole catalysis of **P2**, **P1**, and MTBD for the copolymerization exhibited high copolymer selectivity (≥96%), but low TOFs (1–4 h$^{-1}$, entries 3–5 in Table 1). Concurrently, high reaction temperatures (≥70 °C) or the use of TBD (at 20 °C) only afforded cyclic products (entry 6 in Supplementary Table 2; entry 10 in Supplementary Table 1). PPMTCs obtained by these organic bases had $M_n$s of 29.4–46.0 kg/mol and PDIs of 1.20–1.58. This result exclusively suggested that the COS/PO copolymerization could be performed in an anionic manner.

The activity and selectivity of the above LB-catalyzed COS/PO copolymerization were clearly enhanced by introducing TU-1 (entries 9–12 in Table 1), while TU-1 did not catalyze the reaction alone (entry 11 in Supplementary Table 1). For example, **P4**/TU-1 and **P2**/TU-1 pairs afforded improved TOFs (41 and 24 h$^{-1}$), copolymer selectivity (97/3 and >99) even in high monomer/pair feed ratios of 2000/1/1 and 1000/1/1, respectively, (entries 9–10, Table 1). Concurrently, COS/PO copolymerization via the catalysis of DBU/TU-1 or MTBD/TU-1 pairs led to a slight improvement of TOF (4–7 h$^{-1}$) and the copolymer selectivity (93/7 and 95/5) while the **P1**/TU-1 pair had low TOF (1 h$^{-1}$), but perfect copolymer selectivity (>99) (entry 5, Supplementary

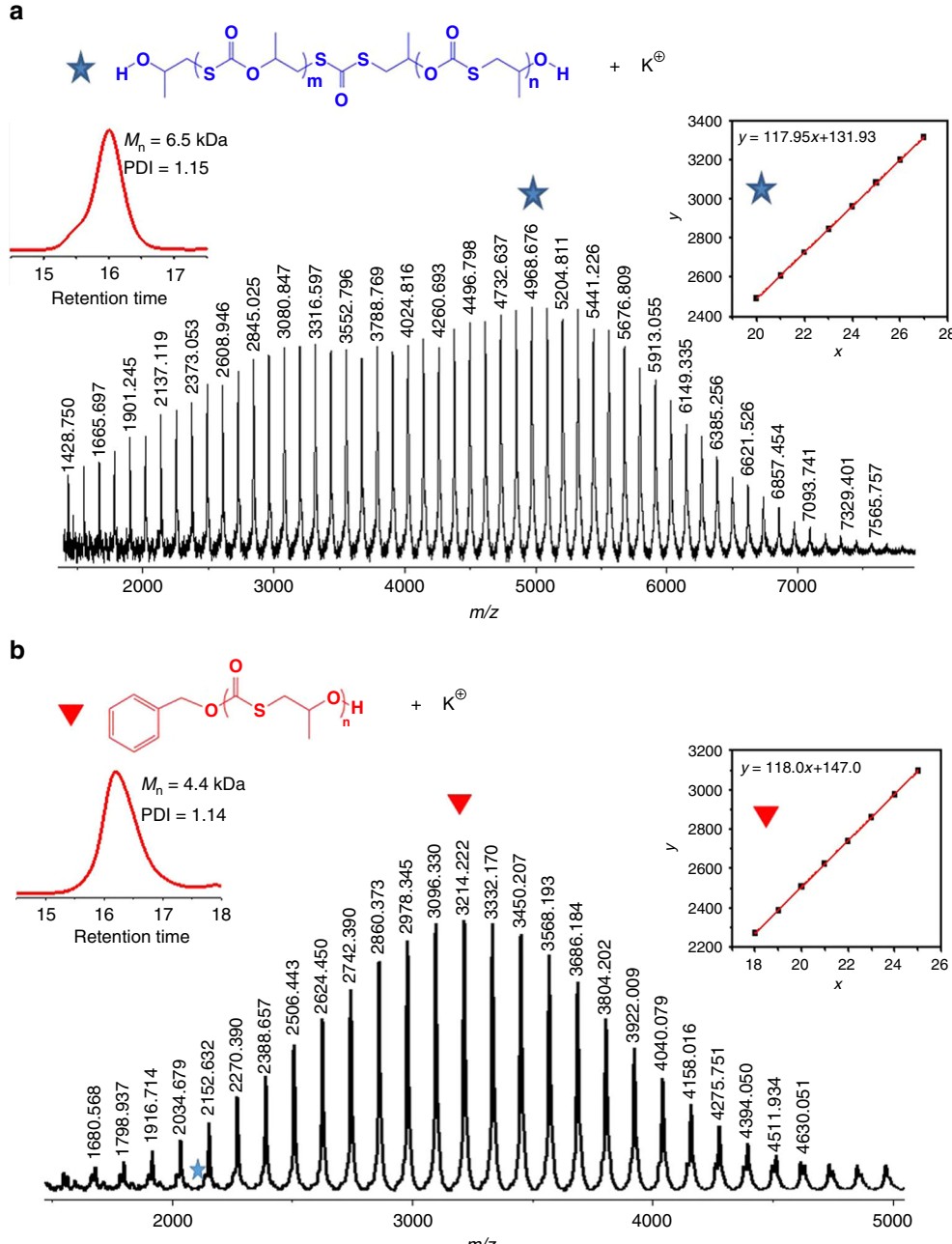

**Fig. 2** Analysis of PPTMCs by MALDI-TOF MS spectra. GPC curves and $M_n$ of PPTMCs and $M_n$ vs. repeated unit number plots are inserted. **a** PPTMC was synthesized without BnOH, afforded α-OH, ω-OH-terminated [H + (PO + COS)$_{m+n}$ + (PS + COS) + PO + OH + K$^+$] copolymer with one dithiocarbonate unit ($M_n$: 6.5 kg/mol; PDI: 1.15); **b** PPTMC was synthesized with BnOH, afforded α-OBn, ω-OH copolymer-terminated [BnO + (PO + COS)$_n$ + H + K$^+$] ($M_n$: 4.4 kg/mol; PDI: 1.14). Reaction conditions: PO/DBU/TU-1(BnOH) = 100/1/1(/1), 25 °C, 3.5 h

Table 1). Of note, the use of TU-1 caused a slight decrease of $M_n$ (30.2–36.1 kg/mol) and narrower PDIs of 1.13–1.16, suggesting the generation of larger amounts of active centers when TU-1 was introduced. Of special interest, MTBD/TU-1 pair catalysis provided a copolymer with a remarkably high $M_n$ of 98.4 kg/mol with PDI of 1.14 after 72 h reaction (entry 13, Table 1), which can be ascribed to a longer lifespan of the active species.

The addition of benzyl alcohol (BnOH) to the COS/PO copolymerization catalyzed by either single LB (entries 6–8 in Table 1, entries 3, 6, and 8 in Supplementary Table 1) or TU-1/LB pairs (entries 14–18 in Table 1, entries 2, 4, 7, and 9 in Supplementary Table 1) led to a dramatic improvement of TOF

without the sacrifice of the copolymer selectivity, even in a rather low catalyst loading. For example, adding equimolar BnOH to **P4** and **P4**/TU-1-catalyzed copolymerizations presented improved TOFs of 55 and 75 h$^{-1}$, respectively, (entries 7 and 14 in Table 1). In addition, high loading of TU-1 and BnOH (**P4**/TU-1/BnOH = 1/5/5) led to a dramatic increase of TOF to 112 h$^{-1}$ (entry 15 in Table 1). Introducing equimolar BnOH to the DBU/TU-1 pair led to a clear promotion of TOF (22 h$^{-1}$) even in a low PO/DBU/TU-1 feed ratio of 2000/1/1(entry 17 in Table 1). In contrast, no copolymers were afforded in the absence of BnOH in PO/DBU feed ratio of 2000/1 under the same reaction conditions (entry 12 in Supplementary Table 1). Actually, LB could activate BnOH

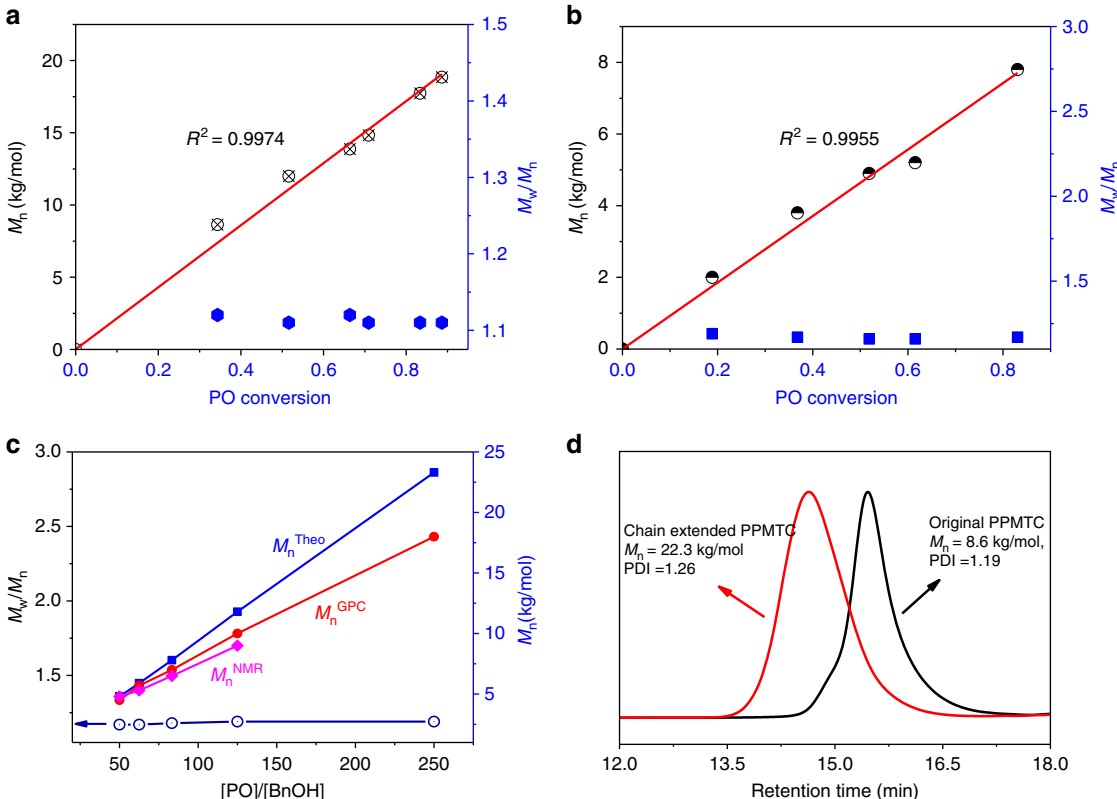

**Fig. 3** Living character of Lewis pair-catalyzed COS/PO copolymerization. **a** Linear plots of $M_n$ of copolymer vs. PO conversion in the absence of BnOH, [DBU] = [TU-1] = 0.2 M, 25 °C; **b** Linear plots of $M_n$ of copolymer vs. PO conversion in the presence of BnOH ([BnOH] = 0.08 M), [DBU] = [TU-1] = 0.2 M, 25 °C; **c** the effect of the [PO]/[BnOH] ratio on $M_n$s of the COS/PO copolymers [calculated value (blue line), and determined by GPC (red line) and NMR (purple line)]; **d** GPC traces of the PPTMCs before and after chain extension

rapidly through H-bond interaction to form alkoxide anion, which initiate the copolymerization[28–31]. Hence, the stronger the basicity of the LB, the higher TOFs of the COS/PO copolymerization. This was evident by the TOF deceasing with the basicity orders of **P4** > **P2** > DBU ≥ MTBD > **P1**. Being totally different from the (salen)CrCl complexes and TEB/LB pairs[9,15], the TU-1/LB pairs could catalyze COS/PO copolymerization in a low COS/PO feed ratio of 1.2/1, even in a feed ratio of 1.05/1 (entries 1–2, Supplementary Table 1). Since COS did not homopolymerize, the copolymerization stopped once PO was nearly completely consumed. The resultant copolymers had 100% of alternating degree (Supplementary Figs. 3–11), indicative of much faster COS activation than successive PO enchainment. O/S ER was also effectively suppressed for all samples in Table 1 and Supplementary Table 1. Moreover, the control experiment ruled out the possible route of ROP of the cyclic monothiocarbonate formed (Supplementary Fig. 12).

**Chain microstructures of COS/PO copolymers**. The perfectly alternating structure and regioregularity of the PPMTCs were also revealed by MALDI-TOF MS spectroscopy (Fig. 2). Figure 2a showed one distribution of α-OH, ω-OH-terminated [H + (PO + COS)$_{m + n}$ + (PS + COS) + PO + OH + K$^+$] copolymer, i.e., a single PPTMC with two –OH end groups and one dithiocarbonate unit ($M_n$: 6.5 kg/mol; PDI: 1.15). Furthermore, high-resolution $^1$H($^{13}$C) NMR spectra (Supplementary Fig. 13a, b) revealed that the copolymer contained two secondary –OH end groups with minimal regio-defect (one dithiocarbonate unit)[40]. These results were indicative of inclusively regioselective attack of the sulfur anion to the CH$_2$ site of PO, and thus the sole

production of the T–H diad via the organocatalysis[40]. In the presence of BnOH, only one distribution of α-OBn, ω-OH copolymer-terminated [BnO + (PO + COS)$_n$ + H + K$^+$] (Fig. 2b) with >99% T–H diad content (Supplementary Fig. 13c, d) was obtained without dithiocarbonate unit ($M_n$: 4.4 kg/mol; PDI: 1.14), meant that BnOH was a very efficient initiator and depressed the production of the dithiocarbonate unit.

**Living COS/PO copolymerization catalyzed by Lewis pairs**. Remarkably, the TU/LB pair catalysis allowed for the copolymerization exhibiting the living features. Take the TU-1/DBU pair-catalyzed COS/PO copolymerization with or without using BnOH (Fig. 3a, b) as instances: linear increase of $M_n$ with PO conversion, narrow PDIs (1.11–1.12, 1.16–1.19) to high PO conversion (89% and 83%, respectively). Simultaneously, the decay in monomer concentration follows zero-order kinetics under various loading of the TU-1/DBU pair in the presence or absence of BnOH or using TU-1/**P2** pair without adding BnOH (Supplementary Fig. 15). In addition, the determined molecular weights by GPC and NMR (i.e.: $M_n^{GPC}$ and $M_n^{NMR}$) were in well agreement with the calculated $M_n^{Theo}$ that increased with the increase of the [PO]/[BnOH] molar ratio in a good linear fashion (Fig. 3c, Supplementary Table 3, and Figs. 17 and 18) while keeping narrow PDIs (1.17–1.19). This result confirmed that $M_n$s of the resultant copolymers could be tuned by changing the [PO]/[BnOH] feeding ratio in the presence of TU-1/DBU pair. Of interest, further investigation showed that the use of exogenous water (even relatively large amounts, no BnOH added) could also effectively initiate the copolymerization and control the molecular weights of the resultant copolymers without changing the chain

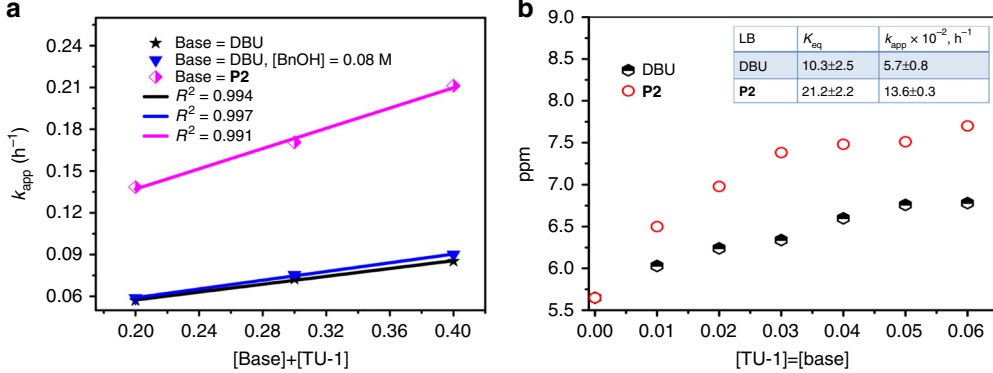

**Fig. 4** The determination of the apparent rate constant ($k_{app}$) and the binding constant of TU-1/base pairs. **a** The plot of the $k_{app}$ vs. [Base] + [TU-1], in neat PO (14.5 M, 14.5 mmol), [TU-1] = [base] = 0.1, 0.15, and 0.2 M; **b** The plot of the chemical shifts of two binding systems with the concentration of TU-1 and base, [base] = [TU-1] = 0.01–0.06 M in CDCl$_3$. $K_{eq}$ is the binding constant (at 297 K), and $k_{app}$ is the apparent rate constant. Errors of $k_{app}$ and $K_{eq}$ are exactly the errors in the slope of the line, determined from linear regression

microstructure (Supplementary Table 4 and Figs. 20–22). As a result, such TU/LB pairs are robust for the copolymerization under mild conditions.

We further explored the chain extension reaction via tandem synthesis (Fig. 3d and Supplementary Fig. 23). PO and COS were copolymerized firstly using TU-1/DBU pair without using BnOH ([PO]: [COS]: [TU-1]: [DBU] = 50: 60: 1: 1) at 25 °C for 10 h. After slowly venting the unreacted COS, PO was totally consumed according to the $^1$H NMR spectra, and the resultant PPTMC exhibited unimodal peak with a $M_n$ of 8.6 kg/mol and a PDI of 1.19. Then, 1.0 ml PO and 1.18 g COS (COS: PO = 1.2: 1) were added into the reactor and sealed for another 20 h reaction at 25 °C. PO conversion was 88% according to $^1$H NMR spectra. The resultant copolymer had a $M_n$ of 22.3 kg/mol with a PDI of 1.26, as revealed by that GPC curve shifted overall to high molecular weight. The result of the chain extension reaction also confirmed the living mode of TU-1/DBU pair-catalyzed COS/PO copolymerization.

**Kinetic and mechanistic study**. The cooperative catalysis of TU-1/LB pair for the COS/PO copolymerization was firmly supported by the kinetic studies (Fig. 4a). The apparent rate constant ($k_{app}$) obtained from the slopes of the best-fit lines to the plots of PO conversion vs. time, is well proportional to [TU-1] + [DBU] in the absence or presence of BnOH, suggesting that the TU-1/DBU pair behaved as a discreet catalyst species [(a) in Fig. 6]. As expected, $k_{app}$ of the TU-1/**P2** pair was $13.6 ± 0.3*10^{-2}$ h$^{-1}$ and higher than that of TU-1/DBU pair ($5.7 ± 0.8*10^{-2}$ h$^{-1}$) for the copolymerization under the same conditions due to the stronger basicity of **P2**. This result is in agreement with the binding constant via $^1$H NMR dilution (see Methods, Supplementary Fig. 24) that was 10.3 ± 2.5 for TU-1/DBU pair and 21.2 ± 2.2 for TU-1/**P2** pair in equilibrium in CDCl$_3$ at 25 °C (Fig. 4b). Because the H-bond interaction could be weakened by elevating the temperature[30], the TU-1/DBU pair catalysis for COS/PO copolymerization at high temperatures (≥55 °C) produced considerable amounts of the cyclic products (Supplementary Table 2), which is consistent with the catalysis involving only the bases for the coupling reaction of COS with PO[41].

We have further studied the combining capabilities of TU-1, DBU, COS, and PO in CDCl$_3$ using $^1$H NMR spectra (Fig. 5 and Supplementary Fig. 25). In a high concentration of TU-1 and DBU (0.5 M), the H-bond interaction was clearly revealed by the proton signal of NH group (5.65 ppm) of TU-1 disappearing while the six protons (NCH$_2$) of DBU became chemically

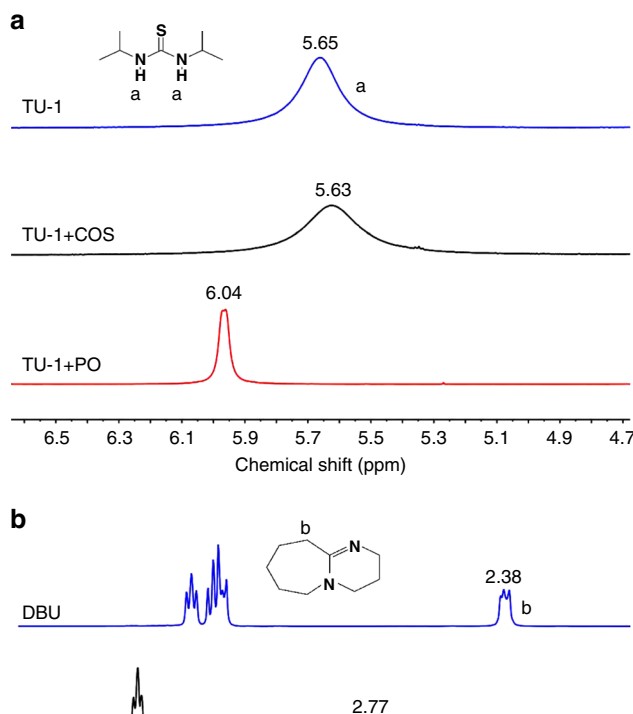

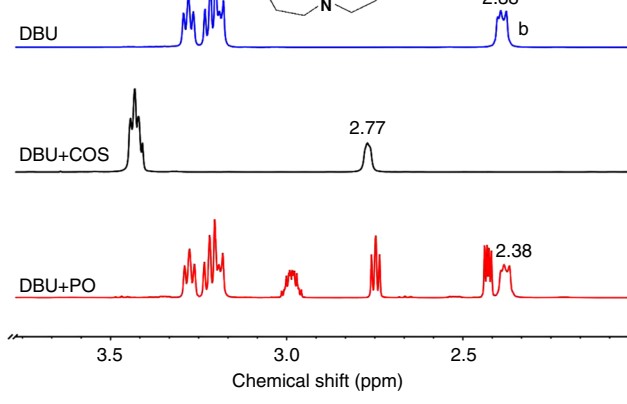

**Fig. 5** The specific recognition of TU-1 to PO and DBU to COS in TU-1/DBU pair via hydrogen-bonding interaction. **a** $^1$H NMR spectra of TU-1, TU-1/COS (1/excess), TU-1/PO (1/60), **b** $^1$H NMR spectra of DBU, DBU/COS (1/excess), DBU/ PO (1/1), 0.5 M TU-1 (DBU) in CDCl$_3$

equivalent (Supplementary Fig. 25). Of interest, TU-1 was shown to solely activate PO and did not interact with COS in CDCl$_3$ (Fig. 5a). Conversely, DBU could only activate COS owing to a clear deshielding effect of protons of NCH$_2$ of DBU (from 2.38 to 2.77 ppm) and appearance of chemical equivalent protons of

**Fig. 6** Proposed anionic copolymerization mechanism. Chain initiation and growth routes are presented upon the cooperative catalysis of TU-1/DBU pair in the presence of BnOH and H$_2$O. Species (*a*)–(*d*) were revealed by Fig. 5 and Supplementary Figs. 24–26; copolymer distributions (*e*) and (*f*) were evidenced by Fig. 2

NCH$_2$ of DBU rather than activate PO (Fig. 5b). Such supramolecular specific recognition of TU-1 to PO and DBU to COS in TU-1/DBU pair promoted the copolymerization cooperatively.

The introduction of BnOH into the polymerization system led to the prior formation of the BnOC(=O)S$^-$…DBUH$^+$ owing to the deprotonation of BnOH by DBU, generated species (*b*) and (*c*) in Fig. 6, as revealed by $^1$H NMR spectra (Supplementary Fig. 26)[42]. Since PO was activated by TU-1 through H-bond [species (*d*) in Fig. 6, revealed by Fig. 5b], the chain growth could be accelerated (Table 1). On the other hand, in the absence of BnOH, trace water (i.e., R=H, Fig. 6) could also initiate the copolymerization via the similar activation route of BnOH by TU/LB pair. Such H$_2$O initiation led to the formation of the end –S(O=C)–OH group, which was thermodynamically unstable and thus converted to –SH group via decarboxylation process[6,40]. Since the end –SH group has the stronger acidity than –OH group, it was rapidly deprotonated, generating bifunctional initiator for further chain growth, a copolymer with two end secondary –OH groups and one dithiocarbonate unit (*f*) was produced, which was clearly revealed by Fig. 2a (Supplementary Figs. 13 and 22).

**The impact of Lewis pairs on COS/epoxides copolymerizations.** Different Lewis pairs including other thioureas were also investigated for the copolymerization of COS with several epoxides. Two other thioureas, TU-2 and TU-3 were synthesized[43] and successfully utilized for the COS/PO copolymerization (entries 1–4 in Table 2). H-bond interactions of both TU-2 and TU-3 with DBU are similar to that of the DBU/TU-1 pair (Supplementary Fig. 28). The TU-3/DBU pair was more active than TU-1 (TU-2)/DBU pairs for the copolymerization, but with a slightly lower copolymer selectivity of 93%. Both TU-2 and TU-3 paired with **P2** showed the same TOFs (29 h$^{-1}$, entries 3–4, Table 2) for the COS/PO copolymerization. Totally, TU-3 with more electron-withdrawing group resulted in the production of slightly more amounts of cyclic products. In addition, two epoxides, glycidyl phenyl ether (PGE) and cyclohexene oxide (CHO) were copolymerized with COS by using **P2**/TU-1, **P4**/TU-1 pairs, affording fully alternating copolymers (entries 5–8, Table 2). Thereof, the COS/PGE copolymerization were fully regioselective with T–H diad content of >99% (Supplementary Fig. 29), while COS/CHO copolymerization could proceed at 40 °C, afforded well-defined copolymer with perfect alternating degree and >99% copolymer selectivity (Supplementary Fig. 30). These results illustrate the use of several low-cost thioureas for the copolymerization of COS and epoxides with different structures.

**Discussion**
We have described the synthesis of perfectly alternating and regioregular poly(monothiocarbonate)s from COS and epoxides by employing thioureas and organic bases under mild conditions. Of importance, the use of thioureas and BnOH led to living/controlled COS/epoxide copolymerization, improved catalytic activity and copolymer selectivity than previous systems. One of the key features of such metal-free catalyst systems is that it can

**Table 2 The copolymerization of COS with various epoxides by using various TU/LB pairs**

| Entry[a] | Epoxide | TU/LB | Epoxide: LB: TU-1 | TOF (h$^{-1}$)[b] | Copolymer selectivity[c] | Alternating degree (%)[c] | T–H diad content (%)[d] | O/S ER product[d] | $M_n$ (kg/mol)[e] | PDI[e] |
|---|---|---|---|---|---|---|---|---|---|---|
| 1 | PO | TU-2/DBU | 500:1:1 | 6 | 97/3 | 100 | >99 | N.F. | 21.9 | 1.15 |
| 2 | PO | TU-3/DBU | 500:1:1 | 10 | 93/7 | 100 | >99 | N.F. | 21.9 | 1.18 |
| 3 | PO | TU-2/**P2** | 1000:1:1 | 29 | 98/2 | 100 | >99 | N.F. | 19.8 | 1.10 |
| 4 | PO | TU-3/**P2** | 1000:1:1 | 29 | 96/4 | 100 | >99 | N.F. | 32.2 | 1.20 |
| 5 | PGE | TU-1/**P2** | 1000:1:1 | 22 | 96/4 | 100 | >99 | N.F. | 29.3 | 1.26 |
| 6 | PGE | TU-1/**P4** | 4000:1:1 | 52 | 96/4 | 100 | >99 | N.F. | 17.5 | 1.18 |
| 7[f] | CHO | TU-1/**P2** | 250:1:1 | 2 | >99 | 100 | — | N.F. | 15.8 | 1.19 |
| 8[f] | CHO | TU-1/**P4** | 250:1:1 | 5 | >99 | 100 | — | N.F. | 13.4 | 1.16 |

Representative NMR spectra are in Supplementary Figs. 29, 30
*PGE* glycidyl phenyl ether, *CHO* cyclohexene oxide
[a–e]Reaction conditions and characterization methods were the same with Table 1
[f]40 °C.

be applied to a variety of epoxides. Due to the ease of tailoring the molecular structures of these organic thioureas and bases, this strategy is a promising alternative to get metal-free well-defined sulfur-containing polymers with high activity and selectivity. Our ongoing efforts are to seek a better understanding of the mechanistic aspects of such catalytic processes, and to develop chiral TU/LB pairs for the copolymerization.

## Methods

**Materials**. Propylene oxide (PO), glycidyl phenyl ether (PGE), and cyclohexene oxide (CHO) were purified by distillation after stirring with calcium hydride for 3 days. 1,3-Diisopropyl-2-thiourea (TU-1) was purchased from Sigma Aldrich and sublimeed before use. 1-cyclohexyl-3-phenylthiourea (TU-2) and 1-[3,5-bis(trifluoromethyl) phenyl]-3-cyclohexylthiourea (TU-3) were synthesized according to literature[43]. $^1$H NMR spectra of TU-2 and TU-3 are shown in Supplementary Fig. 27. *Tert*-butylimino-tris(dimethylamino) phosphorene ($^t$Bu-P$_1$, **P1**, 97%), 1-*tert*-Butyl-2,2,4,4,4-pentakis(dimethylamino)-2λ$^5$, 4λ$^5$-catenadi (phosphazene) ($^t$Bu-P$_2$, **P2**, ~2.0 M in THF) and 1-*tert*-butyl-4,4,4-tris(dimethylamino)-2,2-bis [tris(dimethylamino)- phosphoranylidenamino]-2λ$^5$, 4λ$^5$-catenadi(phosphazene) ($^t$Bu-P$_4$, **P4**, ~0.8 M in hexane) were purchased from Sigma and used directly. 1,5,7-Triazabicyclo[4,4,0]dec-5-ene (TBD) was purchased from Aldrich Chemical Co, which were purified by dissolving in toluene over CaH$_2$, filtering after stirring overnight, and removing the solvent. Benzyl alcohol (BnOH), *N*-methyl-1,5,7-triazabicyclododecene (MTBD) and 1,8-diazabicyclo[5,4,0]undec-7-ene (DBU) were purchased from Alfa Aesar Chemical Co. and Aldrich Chemical Co., respectively, which were purified by distillation over distillation over CaH$_2$ and stored in an inert gas (N$_2$)-filled glove box. Sodium hydride (95%) was purchased from Sigma and used directly. Carbonyl sulfide (COS) (99.9%) was purchased from the APK (shanghai) Gas Company LTD and used as received.

**Representative procedure for copolymerization reactions**. All polymerizations were carried out in glove box under N$_2$ atmosphere unless otherwise specified. A 10-ml autoclave with magnetic stirrer was first dried in an oven at 110 °C overnight, then immediately placed into the glove box chamber. After keeping under vacuum for 1–2 h, the reaction vessel was put into the glove box under nitrogen atmosphere. The copolymerization of COS with PO described below is taken from entry 17 in Table 1 as an example. TU-1 (1.4 mg, 0.007 mmol) was added to the reactor firstly. PO (1.0 ml, 14 mmol) was then carefully added into the vessel. Afterwards, DBU (1.05 μl, 0.007 mmol) and BnOH (0.75 μl, 0.007 mmol) were added into the reactor, respectively. The reactor was sealed and then taken out for charging with the set amounts of COS. The copolymerization was performed at 25 °C for 24 h. Then, the reactor was cooled in ice-water bath, and the unreacted COS was slowly vented. An aliquot was taken from the resulting crude product for the determination of the PO conversion and the molar ratio of copolymer/cyclic products by $^1$H NMR spectrum. Traces of acetic acid were then added for $^1$H NMR spectrum, in order to prevent degradation of the crude product. Next, the crude product was quenched with HCl in ethanol (1 mol/l). The crude product was dissolved with CH$_2$Cl$_2$ and then precipitated in cold methanol. The product was collected by centrifugation and dried in vacuum at 40 °C until a constant weight.

**Characterization**. $^1$H and $^{13}$C NMR spectra were performed on a Bruker Advance DMX 400 MHz or 600 MHz spectrometer. And chemical shift values were referenced to TMS at 0 ppm for $^1$H NMR and $^{13}$C NMR. The number-average molecular weight ($M_n$) and molecular weight distribution ($Đ = M_w/M_n$) of the resultant copolymers were determined with a PL-GPC220 chromatograph (Polymer Laboratories) equipped with an HP 1100 pump from Agilent Technologies. The

GPC columns were eluted with THF with 1.0 ml/min at 40 °C. The sample concentration was 0.4 wt. %, and the injection volume was 50 μl. Calibration was performed using monodisperse polystyrene standards. Matrix-assisted laser desorption/ionization time-of-flight (MALDI-TOF) mass spectrometric measurements were performed on a Waters MALDI Micro MX mass spectrometer, equipped with a nitrogen laser delivering 3 ns laser pulses at 337 nm. Dithranol (97%, Alfa), was used as the matrix. CH$_3$COOK ($\geq$98%, Aladdin) was added for ion formation.

**Binding constant studies**. Equations used for binding studies[44]:

$$\text{For dilution}: \frac{\delta_0 - \delta_i}{[A]_0} = -2K_{eq}(\delta_0 - \delta_i) + K_{eq}(\delta_0 - \delta_\infty) \quad (1)$$

where: $\delta_0$ is the chemical shift of the *ortho*-protons of pure TU-1; $[A]_0$ is concentration of TU-1 or a base; $\delta_i$ is the chemical shift of the *ortho*-protons of TU-1 in the solution when [TU-1] = [base] = $[A]_0$; $\delta_\infty$ is the chemical shift *ortho*-protons of the "pure complex" TU-1 with a base, $(\delta_0 - \delta_\infty)$ is a constant; $K_{eq}$ is the binding constant between TU-1 and a Base.

Following a similar procedure reported by Matthew K. Kiesewetter[29], NMR dilution experiments were carried in CDCl$_3$ with the concentration of [TU-1] = [base] varied from 0.01 M to 0.06 M in CDCl$_3$. DBU and **P2** were employed as bases respectively. The $^1$H NMR spectra was shown in Supplementary Fig. 24. The binding constants ($K_{eq}$) were determined from the slope of the linear forms of the binding equation (above). And the error in $K_{eq}$ is exactly the error in the slope of the line, which can be determined from linear regression.

**Calculation of copolymer selectivity, PO conversion and TOF**. Copolymer selectivity and PO conversion were calculated based on the $^1$H NMR spectrum of the crude product. Taking entry 11 in Table 1 as an example, spectrum of the crude product was showed in Supplementary Fig. 35. Protons with chemical shifts at 4.80, 3.52, 3.26 and 1.51 ppm belong to the methenyl (**d**), methene (**e**), and methyl (**f**) groups, respectively, the peaks at 5.16, 3.08, and 1.36 ppm [also seen in Supplementary Fig. 5, a purified PPTMC] belong to the methenyl (**a**), methene (**b**), and methyl (**c**) groups in the copolymer. And the area ratio of these two parts was taken as the copolymer selectivity. The corresponding peaks in 3.02, 2.77, and 2.46, 1.32 ppm belong to methenyl (**h**), methene (**g**) and methyl (**i**) in PO which was not consumed. On the base of $^1$H NMR spectra, we have:

$$\text{Copolymer selectivity} = \frac{A_{5.16}}{A_{5.16} + A_{4.82}} \times 100\% \quad (2)$$

$$\text{PO conversion} = \left(1 - \frac{A_{2.77}}{A_{5.16} + A_{4.82} + A_{2.77}}\right) \times 100\% \quad (3)$$

Thus,

$$\text{TOF (h}^{-1}) = \text{PO conversion} \times \frac{[\text{PO}]/[\text{LB}]}{t} \quad (4)$$

**Data availability**. The authors declare that the data supporting this study are available within the paper and its Supplementary Information File. All other data is available from the authors upon reasonable request.

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

## Acknowledgements

This work was supported by the National Science Foundation of the People's Republic of China (no. 21774108) and the Distinguished Young Investigator Fund of Zhejiang Province (LR16B040001). We thank Prof. Jun-Peng Zhao (South China University of Technology, China) and Prof. Donald J. Darensbourg (Texas A&M University, USA) for discussion and suggestions.

## Author contributions

X.-H.Z. conceived, designed and directed the investigations, and revised the manuscript. C.-J.Z. carried out all experiments and analyses, contributed much for the sole catalysis of organic bases for the copolymerization and wrote the draft. H.-L.W. and J.-L.Y. carried

out some NMR, MALDI-TOF-MS characterizations and analyses, and discussed with Y. L. for mechanism and kinetic study.

## Additional information

**Competing interests:** The authors declare no competing interests.

