## [Peer Review File · Nature Communications]

Reviewers' comments:

Reviewer #1 (Remarks to the Author):

The manuscript entitled "Precise synthesis of sulfur-containing polymers via supramolecular hydrogen-bonded pairs with high activity" by Zhang et al. describes the alternating copolymerization of carbonyl sulfide and various epoxides using a dual catalytic system, where both catalysts are organic small molecules. The catalytic system results in polymers of moderate molecular weight and the reactions are optimized to achieve impressive selectivity. The authors uncover many mechanistic details about this polymerization system through careful physical-organic studies, which is a great benefit to the reader. Overall, the scholarly presentation of the manuscript is very good.

Upon reading this manuscript, it's apparent that it would be better suited as a full paper. Due to space constraints, the authors have to refer to the supporting information for a significant number of very important pieces of data. That said, this manuscript being the first report of organocatalytic copolymerization of carbonyl sulfide and epoxides is a significant achievement. I would recommend accepting the article after the addition of a few key pieces of data and clarifications (listed below), but I think effective communication of these results would ultimately be better served as a full paper in a journal of similar impact.

Much of the justification is that "metal contaminants impede applications." The authors never explain why this is the case. Many commercial materials are made through metal catalysis. A better justification with supporting citations would strengthen the manuscript.

Although the authors clearly demonstrate the ability to achieve moderate molecular weights, and even higher molecular weights in some cases, there is little discussion about the ability to control molecular weight. If benzyl alcohol is truly acting as an initiator, then monomer to initiator ratio should be able to control molecular weight. However, there is no systematic exploration of this in the manuscript. Being able to target molecular weight would be a significant benefit and deserves exploration.

In figure 2, a lower-case k should be used for rate in the label for the y-axis.

Based on the mechanistic study, I'm do not agree with the title. The polymerization is not catalyzed by "supramolecular hydrogen-bonded pairs". The two catalysts activate different reagents and are not hydrogen bonded as pairs during the reaction. This is better described as an example of cooperative catalysis.

Lastly, why did DBU alone work for you but not for others in the past? What was different? This requires further explanation in the text.

Reviewer #2 (Remarks to the Author):

This is a very solid manuscript on the organocatalytic ring-opening polymerization of carbonyl sulfide and epoxides. The most contribution is on the use of a pair of Lewis acid and base as metal-free catalysts. The authors have carefully examined different roles of Lewis acids and bases. The results are somehow very surprising: an alternative topological composition. The control on polymerization in both molecular weight and molecular weight distribution is excellent. The data

supported most claims. Although the reviewer believes this work is great, the novelty might be in question. Metal-free anionic ROP of CO₂ and epoxides has been reported (references 7). The control is also great, particularly on the similarity of alternating compositions of two monomers. The organocatalysts in this work are not novel (as reported mostly by Waymouth and Hedrick), though they are commercially available. In addition to this argument, the reviewer also has questions about the proposed mechanisms:

(1) If carbonyl sulfide is faster for activation, what is reactivity of alkoxide anion and sulfide anion, how will the difference lead to alternative structures?

(2) If the alternative structure is highly selective, a non-stoichiometric ratio (one monomer is excessive) would lead to either a gradient composition at the penultimate end of chains or a mixture of homopolymers plus copolymers. A clarification is needed.

(3) Is alternative structure unique in terms of polymer properties in comparison with random copolymers? Is there a way to prepare non-alternative copolymers?

Reviewer #3 (Remarks to the Author):

There is a confusion in the entire manuscript between two distinct notions: catalysis and initiation of polymerization. For instance when benzyl alcohol is used it clearly seen that it initiates polymerization as each chain contains the benzyl alcohol moiety (Fig S1B). But when it is not used what species is responsible for of chain growth? Water cannot be the sole responsible for initiation as argued in the manuscript: indeed the MALDI-TOF MS spectroscopy shows two different populations indicative of at least two types of initiation.

Besides the question of the initiation and thus of that of the overall mechanism - LB is said to catalyze polymerization but can't they initiate chains as well? - there is also the question of the factor that controls the molecular weight of the sample. I couldn't find any correlation between the TOFs mentioned and the molecular weights measured.

Also why in the presence of benzyl alcohol the molecular weight jumps to 100kg.mol? There are like that a number of unanswered questions -in particular those related to the mechanism of polymerization- which make this manuscript certainly unsuitable for publication in Nature Communications.

Replies to Reviewer(s)' Comments

Reviewers' comments:

Reviewer #1 (Remarks to the Author):

The manuscript entitled “Precise synthesis of sulfur-containing polymers via supramolecular hydrogen-bonded pairs with high activity” by Zhang et al. describes the alternating copolymerization of carbonyl sulfide and various epoxides using a dual catalytic system, where both catalysts are organic small molecules. The catalytic system results in polymers of moderate molecular weight and the reactions are optimized to achieve impressive selectivity. The authors uncover many mechanistic details about this polymerization system through careful physical-organic studies, which is a great benefit to the reader. Overall, the scholarly presentation of the manuscript is very good.

Upon reading this manuscript, it's apparent that it would be better suited as a full paper. Due to space constraints, the authors have to refer to the supporting information for a significant number of very important pieces of data. That said, this manuscript being the first report of organocatalytic copolymerization of carbonyl sulfide and epoxides is a significant achievement. I would recommend accepting the article after the addition of a few key pieces of data and clarifications (listed below), but I think effective communication of these results would ultimately be better served as a full paper in a journal of similar impact.

Reply: Thank you very much for your encouraging comments and suggestions. We have revised the manuscript to a full paper according to your suggestions. We have also completed several new experiments for intensive mechanistic study. Results and discussion were added in the revised manuscript. Please see the revised text and supporting information, as well as our replies to you as follows.

Much of the justification is that “metal contaminants impede applications.” The authors never explain why this is the case. Many commercial materials are made through metal catalysis. A better justification with supporting citations would strengthen the manuscript.

Reply: We agree. A suitable justification with corresponding citations is added in the revised manuscript. In page 2, we added “ Unfortunately, metal contaminations in the copolymers are resulted and severely impede their applications in optical, optoelectronic, photochemical or biomedical materials.¹²⁻¹⁴”

Although the authors clearly demonstrate the ability to achieve moderate molecular weights, and even higher molecular weights in some cases, there is little discussion about the ability to control molecular weight. If benzyl alcohol is truly acting as an initiator, then monomer to initiator ratio should be able to control molecular weight. However, there is no systematic exploration of this in the manuscript. Being able to target molecular weight would be a significant benefit and deserves exploration.

Reply: We agree. We have done the experiment on studying the relationship of M_n with the

[PO]/[BnOH] molar ratio (BnOH = benzyl alcohol, 250/1~250/5), as shown in Table R1 and Figure R1 (Table S3 in the supporting information and Figure 2C in the text, respectively). In Figure R1, the determined molecular weights by GPC (M_n^{GPC}) and ^1H NMR (M_n^{NMR}) were well consistent with M_n^{Theo} calculated from the [PO]/[BnOH] molar ratio and PO conversion, and the M_n^{GPC} (M_n^{NMR}) increased with the increase of the [PO]/[BnOH] molar ratio in a good linear fashion.

Table R1. Copolymerization of COS with PO in the presence of different amounts of BnOH.^[a]

Entry	[PO]: [BnOH]	PO conv. (%) ^[b]	Copolymer selec. (%) ^[b]	M_n^{Theo} (kg/mol) ^[c]	M_n^{NMR} (kg/mol) ^[d]	M_n^{GPC} (kg/mol) ^[e]	PDI ^[e]
1	250:1	79	98	23.4	-	18.0	1.19
2	250:2	80	98	11.9	9.0	10.0	1.19
3	250:3	79	97	7.9	6.5	7.0	1.18
4	250:4	80	98	6.0	5.3	5.7	1.17
5	250:5	82	95	4.9	4.8	4.5	1.17

[a] Copolymerizations were performed in a 10 mL autoclave at 25 °C for 24 h, [DBU]/[TU-1]/[PO] = 1/1/250, [COS]/[PO] = 1.2/1 in neat PO.

[b] Determined by the integration of ^1H NMR signals. The copolymer selectivity is the molar ratio of the copolymer/cyclic product.

$$[c] M_n^{\text{theo}} = M_{\text{BnOH}} + 118.15 \times \frac{[\text{PO}]}{[\text{BnOH}]} \times \text{PO conv.}$$

$$[d] M_n^{\text{NMR}} = \frac{A_{5.16}}{A_{4.01}} \times 118.15 + M_{\text{BnOH}}, \text{ calculated from the } ^1\text{H NMR spectroscopy of the copolymers (Figure S16).}$$

[e] GPC in THF at 40 °C, polystyrene standard (Figure S17).

Figure R1. The effect of the [PO]/[BnOH] ratio on M_n s of the COS/PO copolymers (Figure 2C in the revised manuscript).

Figure R2. GPC curves of entries 1-5 in Table R1 (small shoulder peak at high molecular weight was caused by trace water).

The above result confirmed that BnOH initiated the copolymerization effectively and control the molecular weights in the presence of TU-1/DBU pair. The result and discussion have been added in the revised manuscript.

Trace amounts of water in the reaction system could also initiate the copolymerization effectively. As shown in Figure R2 (Figure S17 in supporting information), when small quality of BnOH used (250/1~250/2), a clear shoulder peak at the left hand of GPC peak was observed. It could be ascribed to the initiation of trace water in the reaction system. If so, water could act as an initiator with two functionalities, the chain growth along two directions would occur, affording copolymer with a M_n of ca.2 times of the main peak. Take entry 1 in Table R1 as an example, M_n s of the shoulder and main peaks were 31.5 and 16.9 kg/mol (black curve in Figure R3), respectively. In addition, we had completed a control experiment on the COS/PO copolymerization in the presence of BnOH and 0.4 mol% water (relative to

PO), as shown in Figure R3, the total M_n decreased to 10.6 kg/mol, while the intensity of the shoulder peak increased. M_n of the shoulder peak was 17.0 kg/mol and ca. 2 times of the main peak of 8.0 kg/mol (red curve in Figure R3).

Figure R3. GPC curves (black: entry 1 in Table R1; red: 0.4 mol% H₂O was added under the same conditions with entry 1 in Table R1).

In addition, when large amounts of BnOH used (250/3~250/5), the initiation of H₂O for the copolymerization was dramatically inhibited, so that the shoulder peak at the left hand of GPC peak could be minimized.

We have further investigated the impact of exogenous water on the copolymerization in the absence of BnOH, as shown in Table R2 (Table S4 in supporting information) and Figures R4-R5 (Figures S20-S21 in the supporting information part). As our expectation, all GPC curves (Figure R4) presents single elution peak with narrow PDIs (1.11-1.14). The linear variation of M_n^{GPC} (M_n^{NMR}) with the [PO]/[H₂O] feed ratio is in good agreement with that of M_n^{Theo} and the [PO]/[H₂O] ratio. This result also confirmed that H₂O could initiate the copolymerization of COS with PO in the presence of TU-1/DBU pair. And the use of water (even relatively large amounts) can also control the molecular weights of the resultant polymers.

Moreover, the chain microstructure of the copolymer of entry 3 in Table R2 ([PO]/[H₂O] = 100/4) was also investigated by MALDI-TOF-MS spectroscopy (Figure R6, Figures S22 in supporting information), which displayed one distribution with the same structure as Figure 1A, suggesting that use of water didn't change the microstructure of the copolymer.

We have added the corresponding discussions on the control of the M_n by varying the [PO]/[Initiator] ratio in the revised manuscript (Figure 2C, Figure S21), marked with red color.

Table R2. Copolymerization of COS with PO under various [PO]/[H₂O] ratios.^a

entry	[PO]/[H ₂ O]	PO conv.(%) ^[b]	Copolym. Selec. (%) ^[b]	M_n^{Theo} (kg/mol) ^[c]	M_n^{NMR} (kg/mol) ^[d]	M_n^{GPC} (kg/mol) ^[e]	PDI ^[e]
1	100:2	68	91	4.0	3.8	4.1	1.11
2	100:3	65	90	2.6	2.2	2.4	1.10
3	100:4	64	92	1.9	1.5	1.7	1.13
4	100:5	63	93	1.5	1.3	1.2	1.14

[a] Copolymerizations were performed in a 10 mL autoclave at 25 °C for 24 h, [DBU]/[TU-1]/[PO] = 1/1/250, [COS]/[PO] = 1.2/1 in neat PO.

[b] Determined by the integration of ¹H NMR signals. The copolymer selectivity is the molar ratio of the copolymer/cyclic product.

[c] $M_n^{\text{theo}} = M_{\text{H}_2\text{O}} + 118.15 \times \frac{[\text{PO}]}{[\text{H}_2\text{O}]} \times \text{PO conv.}$

[d] $M_n^{\text{NMR}} = \frac{A_{5.16}}{A_{4.01}} \times 118.15 \times 2$, calculated from the ¹H NMR spectroscopy of the copolymers (Figure S19).

[e] GPC in THF at 40 °C, polystyrene standard (Figure S20).

**Figure R4.** GPC curves of entries 1-4 in Table R2.

Figure R5. The effect of H₂O content on the molecular weights of the COS/PO copolymers (Figure S21 in the supporting information part).

Figure R6. MALDI-TOF MS spectra of PPTMCs in entry 3 in Table R2.

In figure 2, a lower-case k should be used for rate in the label for the y-axis.

Reply: Thank you for your careful reminding, it was corrected in the revised manuscript (new Figure 3).

Based on the mechanistic study, I'm do not agree with the title. The polymerization is not catalyzed by "supramolecular hydrogen-bonded pairs". The two catalysts activate different reagents and are not hydrogen bonded as pairs during the reaction. This is better described as an example of cooperative catalysis.

Reply: We do agree. The title has been revised to be "*Precise Synthesis of Sulfur-Containing Polymers via Cooperative Dual Organocatalysts with High Activity*", which could better describe that "The two catalysts activate different reagents and are not hydrogen bonded as pairs during the reaction". We have also discussed the mechanism in detail in a subsection of

“Kinetic and mechanistic study” in the revised manuscript. Thank you so much for such a good suggestion.

Lastly, why did DBU alone work for you but not for others in the past? What was different? This requires further explanation in the text.

Reply: This is really a good question. Our initial design was expected to use “supramolecular hydrogen-bond” to replace “the coordinated bond” for assisting the copolymerization. The first author, Cheng-Jian Zhang carried out some control experiments for comparing the behaviors of the Lewis pairs. He used high loading of DBU and long reaction time ([PO]/[DBU] feed ratio of 250/1 at 25°C for 24 h) and surprisingly observed the production of copolymers (No copolymers were collected when the [PO]/[DBU] feed ratio was 2000/1 at 25°C for 24 h, entry 12 in Table S1). This result supported that it was a *pure* anionic process.

We rechecked the previous literatures and found that considerable amounts of literatures dealing with CO₂/epoxide copolymerization using organic bases including DBU as the *cocatalyst*, to the best of our knowledge, no single bases (like DBU, **P4** etc.) were reported for successful CO₂/epoxide copolymerization. These organic bases (salts) are considered to be inactive to the copolymerization and thus often used as catalyst promotor (in most literatures, *cocatalyst* often used). Different to CO₂, COS has been proved to be more reactive than CO₂, and the thiocarbonate anion [OC(=O)S⁻] is more nucleophilic [*Acc. Chem. Res.* **2016**, *49*, 2209-2219]. Hence, we succeeded in using single organic base for COS/epoxide copolymerization.

In the revised manuscript, we added an explanation and revised the text for this point as follows: “...Since COS is more reactive than CO₂ and the expected thiocarbonate anion [OC(=O)S⁻] is more nucleophilic than carbonate anion [OC(=O)O⁻],⁶ we performed the sole catalysis of LBs (DBU, MTBD, **P4**, **P2** and **P1**) for COS/PO copolymerization as controls (entries 1-5 in Table 1) for comparatively studying the catalytic performance of the designed Lewis pairs of TU/LB (entries 9-18 in Table 1). Unexpectedly, we observed that...” Thank you.

Your suggestions have a significant contribution to the improvement of this manuscript, many thanks to you again.

Reviewer #2 (Remarks to the Author):

This is a very solid manuscript on the organocatalytic ring-opening polymerization of carbonyl sulfide and epoxides. The most contribution is on the use of a pair of Lewis acid and

base as metal-free catalysts. The authors have carefully examined different roles of Lewis acids and bases. The results are somehow very surprising: an alternative topological composition. The control on polymerization in both molecular weight and molecular weight distribution is excellent. The data supported most claims. Although the reviewer believes this work is great, the novelty might be in question. Metal-free anionic ROP of CO₂ and epoxides has been reported (ref. 7). The control is also great, particularly on the similarity of alternating compositions of two monomers. The organocatalysts in this work are not novel (as reported mostly by Waymouth and Hedrick), though they are commercially available.

Reply: Thank you so much for your encouraging comments and suggestions. With regard to the novelty of this manuscript, here are four points:

Firstly, *TU/LB pairs* was used to the alternating copolymerization for the first time. Indeed, similar TU/LB pairs have been applied in ring-opening polymerization (ROP) of cyclic esters and carbonates (representative examples are seen in Figure R7) [ref. 28-31] since the first report by Waymouth and Hedrick [ref. 28]. As far as we know, NO reports were reported on the alternating copolymerization using the TU/LB pairs. Actually, it is a big challenge to realize fully alternating copolymerization of two monomers, especially for two different types monomers [e.g., COS (CO₂) with epoxide] because the catalyst should accommodate two kinds of growing species and allow chain growth perfectly alternative. Herein, on the hypothesis “that the cooperative catalytic process of Lewis pairs composed of TU and base, undergoing a non-covalent mode to activate and stabilize the alcohol initiator/chain end for controlling the anionic copolymerization”, we succeeded in using the TU/LB pairs for COS/epoxide copolymerization with perfect control.

Figure R7. Overview of ROP induced by a dual-activation (from *Progress in Polymer Science* **2016**, *56*, 64-115)

Secondly, the *TU/LB pairs* (including *TU-1/DBU*, *TU-1/P4*, *TU-1/P2* and *TU-1/P1* and others) are new and different to the previously reported systems (Figure R8). The specialties of the TU/LB pairs are the matching of basicity (for example, we provided the binding constant of TU-1/DBU and TU-1/P2 pairs in Figure 3B) and supramolecular specific interactions with two monomers (for example, we studied the combining capabilities of TU-1, DBU with monomers, Figure 4 and Figures S24-S26 in supporting information). In previously reported ROP systems, TUs with electron-withdrawing groups were often used in association with amines, amidines (Figure R8) [ref. 28-31] and metal alkoxides [*Nature*

Chemistry 2016, 8, 1047-1053.]. Compared with the popular TUs, TU-1 was commercially available and low-cost. Of course, we have also provided some data of the pairs of TU-2 and TU-3 (commonly used in previous reports) with LBs for the copolymerization, which presented similar activity but a little bit of side cyclic products. Our ongoing efforts are to seek a better understanding of the difference between the TUs with electron-withdrawing groups and with electron-donating groups (this is another topic that need more design and synthesis of TUs).

Figure R8. Representative organic polymerization catalysts inducing a dual activation (from *Progress in Polymer Science* **2016**, 56, 64-115).

Thirdly, the *TU/LB* pairs exhibited high activity (TOF_{max} : 112 h^{-1} , close to our previously reported value of 119 h^{-1} using triethylborane/LB system) and excellent control on the *COS/epoxide* copolymerization under very mild conditions: perfectly alternating degree, predictable molecular weights, narrow PDIs and well-defined chain microstructure and end groups. We should admit that Ref.16 is definitely the first example metal-free catalysis for synthesizing CO_2 /epoxide copolymer (I really like this work), while 12-15% polyether produced in entries 5-8 in the Table (Figure R9) for CO_2 /PO copolymerization. And the corresponding TOFs were $7.8\text{-}8.2 \text{ h}^{-1}$ in these examples. The TOF_{max} was 49 h^{-1} in entry 14 (Figure R9) while 17% polyether were produced.

Table 1. Results of PO and CO₂ copolymerization by different initiators in the presence of TEB (2 eq. to the initiator). All polymerizations were carried out in 50 mL autoclaves under 10 atm of CO₂ at 60 °C for 10 hrs with equal volume of THF and PO or under conditions otherwise mentioned

Entry	Initiator	Solvent	DP targeted	Yield ^a (%)	TON ^b	PPC ^c (mol%)	Selectivity (%) ^d	M _{n,theor} ^e	M _{n(GPC)} ^f (10 ³ /PDI)
1	tBuOLi	THF	50	8	—	—	— ^g	—	—
2	tBuONa	THF	50	10	—	76	95	0.5	1.6/1.3
3	tBuOK	THF	50	87	43	94	97	4.5	5.1/1.1
4 ^h	tBuOK	THF	50	5	—	—	97	—	—
5	BnOH+P ₂	THF	50	82	41	85	96	4.2	5.8/1.1
6	BnOH+P ₂	THF	100	82	82	88	97	8.4	11.0/1.1
7	BnOH+P ₄	THF	50	81	40	85	95	4.1	5.7/1.1
8	BnOH+P ₄	THF	100	78	78	85	96	8.0	9.5/1.1
9	NBu ₄ Cl	THF	50	85	42	95	97	4.3	4.8/1.1
10 ⁱ	NBu ₄ Cl	THF	50	11	—	—	— ^g	—	—
11 ^j	—	THF/PO	—	—	—	—	—	—	—
12	NBu ₄ Cl	Bulk	50	79	39	82	95	4.0	4.7/1.1
13	NBu ₄ Cl	Bulk	500	71	355	73	94	36.2	43.0/1.1
14	NBu ₄ Cl	Bulk	1000	49	490	83	87	50.0	40.0/1.1
15	NBu ₄ Cl	THF	500	66	330	94	95	34.0	25.0/1.1
16	NBu ₄ Cl	THF	1000	46	460	92	94	45.0	50.0/1.2
17	PPNCl	THF	500	52	260	94	82	26.0	27.0/1.1
18	PPNCl	THF	1000	35	350	98	85	36.0	37.0/1.2
19	Ph ₄ P ⁺ Cl ⁻	THF	1000	19	190	99	88	20.0	15.0/1.1

^a Calculated by gravimetry. ^b TON = mol_{(PT)_{consumed}} / mol_(initiator). ^c Calculated by ¹H NMR. ^d Calculated from IR spectra. ^e Calculated based on the formula: M_{n,theor} = 102 × (DP_{target}) × (Yield%). ^f Determined by GPC in chloroform with polystyrene standard. ^g Only cyclic carbonate(propylene carbonate) was obtained. ^h TiBA was used instead of TEB. ⁱ No TEB was added. ^j Only TEB was added with same composition of TEB and PO as in entry 9.

Figure R9. Table 1 extracted from *J. Am. Chem. Soc.* **2016**, *138*, 11117-11120 entitled “Metal-Free Alternating Copolymerization of CO₂ with Epoxides: Fulfilling "Green" Synthesis and Activity” by Prof. Gnanou and Feng and coworkers (herein, the copied Table is only for review).

Finally, *the chemistry of COS/epoxide copolymerization is more complicated.* Sorry we had not presented the specialty of COS/epoxide copolymerization in the original manuscript. We added the discussion in the revised manuscript as follows: “... However, anionic copolymerization of COS with epoxides remains unexplored. In contrast with the CO₂/epoxide copolymerization that is often expected to attain fully alternating structure and no production of side cyclic carbonate (i.e. 100% polycarbonate),¹⁹⁻²⁴ the chemistry of COS/epoxide copolymerization is more complicated.⁶ One is the possible occurrence of oxygen/sulfur exchange reactions (O/S ERs), which cause the production of CO₂, and thiirane intermediate, will produce randomly distributed dithiocarbonate and carbonate units in the final copolymer.^{6, 25-27} The other is that the copolymerization of structurally asymmetric COS with a terminated epoxide, will generate four consecutive monothiocarbonate diads, i.e.: head-to-tail (H-T), tail-to-head (T-H), tail-to-tail (T-T), and head-to-head (H-H) diads.⁶ As a result, metal-free catalyst for anionic COS/ epoxide copolymerization should avoid O/S ER and attain highly regioselectivity involved two asymmetric monomers, and be precisely controlled by varying the monomer to initiator ratios under mild condition. ...” in the third paragraph. Actually, before the experiment result came out, honestly, we were not sure whether TU/LB pair could overcome the above difficulties.

Thank you for raising this question, we hope that the above responses could make the novelty appeared.

In addition to this argument, the reviewer also has questions about the proposed mechanisms: (1) If carbonyl sulfide is faster for activation, what is reactivity of alkoxide anion and sulfide anion, how will the difference lead to alternative structures?

Reply: Yes, as illustrated in Scheme R1, the following conditions were fulfilled, allowing the alternating copolymerization to take place: 1) the rate constant of COS addition to the growing alkoxide chain-end (k_{p12}) was much higher than the rate constant of ROP of epoxide (k_{p11}), thus preventing the formation of ether units; 2) the chain ends bounded with sulfide anion could ring-open epoxides ($k_{p21} > 0$), due to the higher nucleophilicity of sulfide anion than the alkoxide anion.

Scheme R1. Possible reactions involved in anionic COS/PO copolymerization ($k_{p12} \gg k_{p11}$).

(2) If the alternative structure is highly selective, a non-stoichiometric ratio (one monomer is excessive) would lead to either a gradient composition at the penultimate end of chains or a mixture of homopolymers plus copolymers. A clarification is needed.

Reply: Yes, the copolymerization is non-stoichiometric with a chain polymerization mechanism. Excess of COS favored the formation of fully alternating structure. In the original manuscript, COS was excess than PO (COS: PO = 1.2 : 1). We have added the clarification “Since COS did not self-polymerize (confirmed by NMR and MALDI-TOF-MS data), the copolymerization stopped once PO was nearly completely consumed” in the revised manuscript (page 7). It is also *living* because continuously adding PO led to the chain extension in the presence of COS, as shown in Figure 3D.

[Redacted]

(3) Is alternative structure unique in terms of polymer properties in comparison with random copolymers? Is there a way to prepare non-alternative copolymers?

Reply: Yes. For the copolymerization of C1 monomers (e.g.: CO₂, COS), we often hope to get a perfectly alternating copolymerization, because the resultant copolymers have more

functionalities (e.g.: carbonate, thiocarbonate), which endowed copolymers unique properties, such as biodegradability, or high transparency. In addition, perfectly alternating copolymerization could also utilize “waste” gas (e.g.: CO₂, COS) as much as possible. For the last question, we have also focused on the preparation of non-alternative copolymers of C1 and epoxide, for example, poly(ether-*co*-carbonate) diols from CO₂ and PO (Macromolecule, 2015 48, 536–544; polymer, 2011, 52, 5494-5502). These polyols are expected to be used in making polyurethanes. For COS-epoxide copolymerization, polyether could be also produced when metal catalysts used (Polymer 2014, 55, 3688-3695). Thank you for raising a new research perspective.

Reviewer #3 (Remarks to the Author):

There is a confusion in the entire manuscript between two distinct notions: catalysis and initiation of polymerization. For instance, when benzyl alcohol is used it clearly seen that it initiates polymerization as each chain contains the benzyl alcohol moiety (Fig S1B). But when it is not used what species is responsible for of chain growth? Water cannot be the sole responsible for initiation as argued in the manuscript: indeed the MALDI-TOF MS spectroscopy shows two different populations indicative of at least two types of initiation.

Reply: Thank you for your good question. When initiator (BnOH) was not added, trace water in the reaction system predominantly initiated the copolymerization. Sorry that the original manuscript did not present detail discussion on this side.

To further explain the structure in Figure 1A, we have further investigated the effect of exogenous water on the copolymerization in the absence of BnOH, as shown in Table R2 (Table S4 in supporting information) and Figures R4-R5 (Figures S20-S21 in the supporting information part). As expected, all GPC curves (Figure R4) presents single elution peak with narrow PDIs (1.11-1.14). The linear variation of M_n^{GPC} (M_n^{NMR}) with the [PO]/[H₂O] feed ratio is in agreement with that of M_n^{Theo} and the [PO]/[H₂O] ratio. This confirmed that H₂O could initiate the copolymerization of COS with PO in the presence of TU-1/DBU pair. In addition, the chain microstructure of the copolymer obtained in the presence of considerable amounts of water (e.g.: entry 3 in Table R2, [PO]/[H₂O] = 100/4) did not changed according the result of MALDI-TOF-MS spectroscopy (Figure R6, Figures S22 in supporting information).

Because the original manuscript is a short communication style, so the result related to the water initiation was a sudden and causes confusion (e.g.: the explanation on Figure 1A was quite simple). Now it was extended to be a full paper according to the suggestions of the reviewer-1 and editor. At the same time, according to your suggestion, a detail discussion on the chain initiation mechanism is proposed based on the experimental data (including newly added data), as shown in Scheme R2.

Scheme R2 (Scheme 2 in the revised manuscript): Proposed chain initiation and growth upon the cooperative catalysis of TU-1/DBU pair in the presence of BnOH and H₂O. Species (1)–(4) were revealed by Figure 4 and Figures S24–26; copolymer distributions (5) and (6) were evidenced by Figure 1.

In Scheme R2, trace amounts of water in the reaction system (R=H), “... in the absence of BnOH, trace water (i.e., R=H, Scheme 2) could also initiate the copolymerization via the similar activation route of BnOH by TU/LB pair. Such H₂O initiation led to the formation of the end -S(O=C)-OH group, which was thermodynamically unstable and thus converted to -SH group via decarboxylation process.^{6,40} Since the end -SH group has the stronger acidity than -OH group, it was rapidly deprotonated, generating bifunctional initiator for further chain growth, a copolymer with two end secondary -OH groups and one dithiocarbonate unit (6) was produced, which was clearly revealed by Figure 1A (Figures S13 and S22).” Therefore, the production of distribution in Figure 1A [species (6) in Scheme 2] can be reasonably explained.

In addition, we had done a control experiment on the COS/PO copolymerization in the presence of BnOH and 0.4 mol% water (relative to PO), as shown in Figure R3, the total M_n decreased to 10.6 kg/mol, while the intensity of the shoulder peak increased. M_n of the

shoulder peak was 17.0 kg/mol and ca. 2 times of the main peak of 8.0 kg/mol (red curve in Figure R3).

Besides the question of the initiation and thus of that of the overall mechanism - LB is said to catalyze polymerization but can't they initiate chains as well? - there is also the question of the factor that controls the molecular weight of the sample. I couldn't find any correlation between the TOFs mentioned and the molecular weights measured.

Reply: Thank you for your question. In the original manuscript, the sentence of “Therefore, adding equimolar of PO and excess COS into DBU/TU-1 pair may form a sulfur anion of DBU-COS⁻, which could attack the activated PO by TU-1 through H-bond, and thus initiate the copolymerization.” is wrong based on the data (**NOT** DBU-COS⁻, need deprotonation step). A detail chain initiation and growth upon cooperative catalysis of TU-1/DBU pair is proposed, as shown in Scheme R2. The role of LB (DBU as an example) is presented. In this protocol, the initiators (BnOH, H₂O) could be effectively deprotonated by DBU, COS was then activated and to form species (3), which could be revealed by ¹H NMR spectrum (Figures S26).

In the revised manuscript, we have also quantitatively investigated the control of the molecular weights of the copolymers according to your suggestions. As shown in Tables R1-R2, Figure R1-R5, the relationship of M_n with the [PO]/[I] feed ratio (I was BnOH, or water) is clearly presented. We have also added the discussion in the revised manuscript.

Figure R11. Linear plots of M_n of copolymer vs. PO conversion when [DBU] = [TU-1] = 0.2 M, [BnOH] = 0.08M, 25°C.

Figure R12. GPC curves of COS/PO copolymers under TU-1/DBU catalysis in the presence of BnOH (small shoulder peak at high molecular weight was caused by trace water).

Moreover, according to your question, we have also investigated the relationship of M_n vs. PO conversion (i.e., TOF), as shown in Figure R11 (Figure 2B in the revised manuscript). It showed that M_n increased linearly with increasing PO conversion in the presence of BnOH, although trace water initiated the copolymerization and caused a small shoulder peak, PDIs are 1.16-1.19 and still narrow (Figure R12, Figure S14 in the supporting information part). This result suggests that this copolymerization can be well controlled.

Also why in the presence of benzyl alcohol the molecular weight jumps to 100kg.mol? There are like that a number of unanswered questions -in particular those related to the mechanism of polymerization- which make this manuscript certainly unsuitable for publication in Nature Communications.

Reply: Thank you for your questions. You may refer to another data in Table 1, entry 13 showed that a copolymer with M_n of 98.4 kg/mol was obtained *in the absence of BnOH* while MTBD was used as LB. This example was used to demonstrate that the TU-1/LB pair has long life span because these organic molecules are not poisoned by trace water or air in most cases. In this example, because quite low loading of TU-1/MTBD pair used without BnOH ([PO]: [MTBD]: [TU-1] = 1000:1:1), and only trace water in the reaction system initiated the copolymerization, high M_n copolymer was thus achieved after a long reaction time (72 h).

We understand that your question may be that the relationship of the dosage of benzyl alcohol with the molecular weight of the final copolymers, we had completed the experiment, and the results are in Table R1 and Figures R1-R3 (responses to reviewer-1). The result disclosed that the impact of the [PO]/[BnOH] feeding ratio on M_n s of the COS/PO copolymers. The determined molecular weights by GPC (M_n^{GPC}) and $^1\text{H NMR}$ (M_n^{NMR}) were well consistent with M_n^{Theo} calculated from the [PO]/[BnOH] molar ratio and PO conversion,

and the M_n^{GPC} (M_n^{NMR}) increased with the increase of the [PO]/[BnOH] molar ratio in a good linear fashion. That is, BnOH could initiate the copolymerization effectively and control the molecular weights in the presence of TU-1/DBU pair. We have added Figure 2B-2C and corresponding discussions according to your suggestions.

Thank you very much for your constructive suggestions especially on the mechanistic study, which indeed contributed much to the improvement of this manuscript.

After this careful revision according to all the comments and suggestions by all reviewers and suggestions, we believe the quality of this manuscript was improved.

REVIEWERS' COMMENTS:

Reviewer #1 (Remarks to the Author):

The manuscript is much improved from its original version. The experiments demonstrating that the system can target specific molecular weights and the kinetic studies are especially impressive. I recommend publication in Nature Communications.

Reviewer #2 (Remarks to the Author):

I applaud the authors have taken very seriously on every comment from each reviewer. They offered clarification, explanations and many more experiments to augment newly revised manuscript. After carefully reading it, I am very satisfied with response toward my comments. Thus, I think this paper is ready for publication in Nature Communications.

Reviewer #3 (Remarks to the Author):

I reviewed the revised version of the manuscript "Precise Synthesis of Sulfur-Containing Polymers via Cooperative Dual Organocatalysts with High Activity" submitted by Prof. Zhang and his coworkers and I overall agree with almost all changes introduced but one major. There is still one major source of confusion in this manuscript: as I read it and it is clearly explained in scheme 2 the mechanism of polymerization is nothing but a mere anionic polymerization in which one of the two monomers –here the epoxide- is activated by thioureas. The use of the term "catalysis" in this context is thus misleading! Even the initiator is the result of the joint activation of an alcohol by a Lewis Base and thioureas. In the entire manuscript the term "catalysis" should thus be eliminated and the clarification made as to a mechanism proceeding anionically and involving monomer activation by thioureas!

Responses to the referees.

Reviewer #1 (Remarks to the Author):

The manuscript is much improved from its original version. The experiments demonstrating that the system can target specific molecular weights and the kinetic studies are especially impressive. I recommend publication in Nature Communications.

Reply: Many thanks for your encouraging recommendation.

Reviewer #2 (Remarks to the Author):

I applaud the authors have taken very seriously on every comment from each reviewer. They offered clarification, explanations and many more experiments to augment newly revised manuscript. After carefully reading it, I am very satisfied with response toward my comments. Thus, I think this paper is ready for publication in Nature Communications.

Reply: Thank you very much for your nice recommendation.

Reviewer #3 (Remarks to the Author):

I reviewed the revised version of the manuscript "Precise Synthesis of Sulfur-Containing Polymers via Cooperative Dual Organocatalysts with High Activity" submitted by Prof. Zhang and his coworkers and I overall agree with almost all changes introduced but one major.

There is still one major source of confusion in this manuscript: as I read it and it is clearly explained in scheme 2 the mechanism of polymerization is nothing but a mere anionic polymerization in which one of the two monomers –here the epoxide- is activated by thioureas. The use of the term "catalysis" in this context is thus misleading! Even the initiator is the result of the joint activation of an alcohol by a Lewis Base and thioureas. In the entire manuscript the term "catalysis" should thus be eliminated and the clarification made as to a mechanism proceeding anionically and involving monomer activation by thioureas!

Reply: Thank you very much for your encouraging comments, and of course we thank you for your concerns on the use of term "catalysis" because your questions guided us to deeply think the proposed mechanism. I have the following reasons that might be convincing.

Firstly, the role of "Thioureas/LB" in the copolymerization is well agreement with the definition of *catalysis*, that is, the increase in the rate of a chemical reaction due to the participation of an additional substance called a catalyst, which is not consumed in the catalyzed reactions and can continue to act repeatedly (Wikipedia). In a published review work, Yann Sarazin and Jean-François Carpentier presented a *true catalyst* (with definition in polymer chemistry) for epoxide-involved polymerization (Discrete Cationic Complexes for Ring-Opening Polymerization Catalysis of Cyclic Esters and Epoxides, Chem. Rev. 2015, 115, 3564-3614, in page 3566-3567). Herein, the Lewis

bases activated the initiator (H₂O or BnOH) and thioureas activated epoxides, making a cooperative dual organocatalytic process (in some informal cases, thioureas/LB pair is called “activator”). We considered that thiourea accelerated the copolymerization process as an additional substance and did not change chemically after the reaction. This meets the definition of the “catalyst”.

Secondly, the term of “organocatalysis” (in the field of polymer chemistry) is usually used to describe the role of the amines (amidines)/thioureas systems, which is developed by Prof. Waymouth and other groups [see refs 28-31. for examples, Dove, A. P.; Pratt, R. C.; Lohmeijer, B. G. G.; Waymouth, R. M.; Hedrick, J. L. *Thiourea-based bifunctional organocatalysis: supramolecular recognition for living polymerization*. *J. Am. Chem. Soc.* **127**, 13798-13799 (2005); Kiesewetter, M. K.; Shin, E. J.; Hedrick, J. L.; Waymouth, R. M. *Organocatalysis: Opportunities and challenges for polymer synthesis*. *Macromolecules* **43**, 2093-2107 (2010); Kazakov, O. I.; Datta, P. P.; Isajani, M.; Kiesewetter, E. T.; Kiesewetter, M. K. Cooperative hydrogen-bond pairing in organocatalytic ring-opening polymerization. *Macromolecules* **47**, 7463-7468 (2014)].

Finally, for a pure anionic polymerization, in most cases, it is indeed a catalytic process that the catalysts are often used to generate propagating anions through activating initiators or monomers.

As a result, we considered that the use of the term “catalysis” is suitable in the text. Thank you again for your supporting even we have different ideas on “catalysis”. I believe such arguments will lead us to approach the truth more closely.